# The Incidence, Impact, and Techniques of Commissural Alignment in Transcatheter Aortic Valve Implantation: A Review

**DOI:** 10.3390/jcm12237369

**Published:** 2023-11-28

**Authors:** Jose G. Paredes-Vazquez, Gabriela Tirado-Conte, Asad Shabbir, Matias Mon-Noboa, Jorge F. Chavez, Ivan Nuñez-Gil, Pilar Jimenez-Quevedo, Eduardo Pozo-Osinalde, Jose Juan Gomez de Diego, Pablo Salinas, Hernan Mejia-Renteria, Fernando Macaya, Jose Alberto de Agustin-Loeches, Nieves Gonzalo, Javier Escaned, Antonio Fernandez-Ortiz, Luis Nombela-Franco

**Affiliations:** Cardiovascular Institute, Hospital Clínico San Carlos, Instituto de Investigación Sanitaria del Hospital Clínico San Carlos (IdISSC), 28040 Madrid, Spain; dr.jose.paredes.vazquez@gmail.com (J.G.P.-V.);

**Keywords:** commissure alignment, TAVI, aortic stenosis, coronary reaccess

## Abstract

In current clinical practice, commissural alignment of the transcatheter heart valve (THV) during transcatheter aortic valve implantation (TAVI) is seldom achieved. Orientation of the THV within the aortic root and the subsequent influence upon leaflet haemodynamic function, coronary blood flow, and ease of access to the coronary ostia are gaining significant interest. Herein, we review the incidence and clinical implications of commissural misalignment in TAVI and offer thorough descriptions of how optimal alignment can be achieved with several different contemporary THV devices.

## 1. Introduction

Transcatheter aortic valve implantation (TAVI) is an established therapeutic option amongst patients with aortic stenosis (AS) with high- and intermediate surgical risk [1]. However, the current trend is to extend the use of TAVI to younger and lower-risk patients with a longer life expectancy [2,3]. This practice represents a commitment to improve the long-term clinical outcomes, specifically, the durability of the transcatheter heart valve (THV) device, whilst being cognisant of a potential need of future interventions in cases of prosthesis dysfunction or in instances where coronary access is needed to treat coronary artery disease. Throughout the last two decades, significant advances in the design of THVs have been achieved with several generations of device development, and innovation has enabled improved procedural, clinical, and haemodynamic outcomes for patients.

Despite this, procedural aspects pertaining to commissural alignment (CA) must be addressed. Orientation of the THV within the aortic root and the subsequent influence upon leaflet haemodynamic function, coronary blood flow, and ease of access to the coronary ostia are gaining significant interest [4,5]. Herein, we offer an in-depth analysis of CA. Firstly, we present principal definitions, followed by thorough descriptions of when, why, and how CA can be performed with different THVs.

## 2. Definitions

Commissural alignment is defined according to the angular relationship between the native and bioprosthetic valve commissures. A mean angle, measured in degrees, between the three native and the three neo-commissures of between 0° to 15° is considered to be an optimal CA.

A mean angle of greater than 15° is termed commissural misalignment (CMA). Categorisation of CMA severity has recently been proposed by the ALIGN-TAVR (Alignment of Transcatheter Aortic-Valve Neo-Commissures) consortium, whereby mild CMA was defined as between 15–30°, moderate as 30–45°, and severe as 45–60° (Figure 1A) [6].

Aortic valve cusp symmetry is the inter-commissural angle of the largest cusp, considering a symmetrical cusp in the range of 120–125°. The severity of cusp asymmetry is sub-categorised into mild (125.1–130.0°), moderate (130.1–135.0°), and severe (>135°) (Figure 1C).

Coronary ostial eccentricity is defined by the angle between the cusp centre and the coronary ostium and is considered centred when it is between 0° and 10°. Mild coronary eccentricity is defined when it is within a range of 10.1–20.0°, moderate between 20.1–30.0°, and severe when >30° (Figure 1D). Moderate-to-severe coronary ostial eccentricity in combination with a degree of CMA can cause coronary overlap (CO), which is defined as an angular distance from the nearest commissural post to a coronary ostium of <15° (Figure 1B), increasing the probability of coronary artery occlusion (CAO), owing to the short distance between the commissure and the coronary ostia in combination with even a minimal degree of CMA. Coronary eccentricity of the right coronary of >24.5°, and of the left coronary of >19°, represents a moderate-to-severe risk of CO despite ideal CA during implantation [7]. This concept has been recently addressed, and coronary alignment in such cases of marked coronary eccentricity is undertaken with a different approach.

Conceptually, optimal coronary alignment is the positioning of one of the neo-commissures at the bisector angle between the coronary artery ostia (Figure 1E). It can be identified in the pre-procedural computed-tomography angiogram (CTA), calculating the projection with the C-arm with both coronary ostia aligned on the right side of the image to implant one of the bioprosthesis posts and overlapping with two posts on the left side, avoiding coronary overlap independent of CA. In this way, the risk of moderate-to-severe CO is reduced considerably from 32% to 5% [7].

## 3. Frequency of Commissural Misalignment

Before the concept of CA became a consideration in TAVI, commissural orientation appeared to be random. CMA incidence varied between 53% to 68% in published evidence amongst different THVs. Fuchs et al. assessed CA between bioprosthetic and native aortic valve leaflets following surgical aortic valve replacement (SAVR) (*n* = 28) and TAVI (*n* = 212), where 96% of SAVR prostheses were aligned and only 22% of TAVI prostheses were optimally aligned. Mild, moderate, and severe CMAs were present in 25%, 22%, and 31% of the patients, respectively [4]. This cohort included similar numbers of different THVs in each group of CMAs. Tang et al. reported a retrospective consecutive cohort of 70 patients to determine the frequency and extent of CO that included patients treated with TAVI, with the balloon-expandable SAPIEN family bioprosthesis (Edwards Lifescience, Irvine, CA, USA) or the CoreValve/Evolut R (Medtronic Inc, Minneapolis, MN, USA). Severe CO occurred in 26.4% of the SAPIEN family and 17.6% in the CoreValve/Evolur R bioprosthesis, with a total frequency of 51.4%, potentially impacting coronary re-access after TAVI [8]. Similar findings were observed by Raschpichler et al., who retrospectively analysed CTAs from the RESOLVE (Assessment of TRanscathetEr and Surgical Aortic BiOprosthetic Valve Thrombosis and Its TrEatment With Anticoagulation) registry, identifying moderate or severe CMA (>30°) in 52.8% (*n* = 324) of patients treated with the SAPIEN 3 bioprosthesis [9]. In addition, Tirado-Conte et al. reported on a cohort of 66 patients with Portico THVs (Abbott Structural Heart, Minneapolis, MN, USA) that moderate or severe CMA occurred in 68% of patients using a delivery catheter insertion with the flush port at 12 o’clock [10].

## 4. Clinical Implications of Commissural Alignment

Since the realisation that CA might influence post-TAVI clinical outcomes, research has been oriented towards five principal concerns: (1) coronary access after TAVI, (2) valve haemodynamics, (3) valve durability, (4) leaflet thrombosis, and (5) aortic valve reinterventions (redo-TAVI) with the possibility of performing upfront coronary protection with leaflet modification techniques. These topics are described in greater detail below.

### 4.1. Coronary Access

Coronary angiography is frequently undertaken in patients with previous TAVI, and coronary access remains an important concern, especially in the setting of acute coronary syndrome (ACS). The interaction between the coronary catheter and the prosthesis varies depending on the type of THV implanted (Figure 2). The feasibility of coronary ostia cannulation was assessed in the RE-ACCESS (Reobtain Coronary Ostia Cannulation Beyond Transcatheter Aortic Valve Stent) study, whereby Barbanti et al. performed pre- and post-TAVI coronary angiography in 300 patients with balloon- and self-expandable prostheses implanted without CA. A total of 23 cases (7.7%) had unsuccessful coronary cannulation (14 (4.7%) in the LCA and 12 (4.0%) in the RCA), with the majority (96%) being in the Evolut group [11]. Semi-selective cannulation was reported in 36 (12%) in the LCA and 95 (31.7%) in the RCA; again, this occurred more frequently in patients treated with Evolut (adjusted OR, 29.6). In the ALIGN-ACCESS (TAVR With Commissural Alignment Followed by Coronary Access) study, Tarantini et al. investigated the impact of CA on the feasibility of coronary access following TAVI in 206 patients treated with SAPIEN 3, Evolut R/Pro, and Acurate Neo. The self-expandable THVs were implanted with successful CA in 85.9% and 88.5%, respectively [12]. Unsuccessful cannulation of the LCA or RCA was observed in only 3% (0 and 5% with SAPIEN 3, 6% with Evolut R/Pro, and 3% with Acurate Neo, *p* < 0.001). Selective coronary cannulation was achieved in 77% of cases (95% with SAPIEN 3, 70% with Evolut R/Pro, 61% with Acurate Neo, *p* < 0.001). The authors reported that non-selective cannulation and unsuccessful cannulation were observed more frequently in non-aligned cases (43% versus 26%, and 11% versus 3%, *p* = 0.01, respectively) exclusively in the group with self-expandable supra-annular THVs. Thus, CA seems to improve the feasibility of coronary access after TAVI, but even with CA techniques, failure to achieve coronary access post-TAVI remains more frequent in THVs with a supra-annular leaflet position [13]. Additional risk factors for unsuccessful coronary cannulation include high THV implantation (cutoff value of −6 mm (OR, 1.7)), small SoV diameter (−0.7% (OR, 1.1)), and a low SoV height (>20 mm (OR, 0.83)) [11,12]. Additionally, the THV cell dimensions, the presence of a covered cell facing the coronary ostium, or the THV skirt upper edge above the ostium (short THV inflow-to-ostium distance), the THV stent frame above the STJ, and the coronary height (mainly in SAPIEN THV) are also associated with challenging coronary re-access (Figure 3) [14,15]. Unfavourable coronary access can be identified in a post-TAVI CTA as a coronary ostium below the outer skirt or in front of the THV commissural posts. This definition was associated with a rate of unsuccessful coronary access of 78% with the Evolut THV and 31% with SAPIEN 3 [16].

Coronary eccentricity is also related to CA and coronary re-access after TAVI. An incidence of moderate-to-severe coronary eccentricity has been reported in ~30%, which represents a higher risk of coronary overlap with the neo-commissure even with adequate CA [7,17]. For this reason, in the presence of significant coronary eccentricity, patient-specific coronary alignment (coronary ostium overlap view or inter-coronary ostium view) might be more useful than the standard CA technique. However, additional clinical evidence and greater standardisation are required to corroborate this concept [17,18].

In summary, coronary access after TAVI depends on patient-specific anatomy, THV characteristics, and implantation technique. Small aortic root, low coronary ostia, and coronary eccentricity are anatomical features limiting post-TAVI coronary access. Self-expanding supra-annular THVs, such as the Evolut or Acurate Neo 2 devices, which are designed to optimise valve haemodynamics, reduce leaflet stress, and improve valve durability, may also interfere with coronary access. Moreover, techniques resulting in higher implantations to avoid conduction disturbances may also hinder coronary re-access, particularly in cases of CMA. The mechanism behind this is the physical barrier between the catheter and the coronary as the THV commissural post or THV skirt is facing towards the coronary ostium; these are the cases where CA seems more relevant to improve the feasibility of coronary access after TAVI. Conversely, THV with intra-annular leaflets such as the SAPIEN 3 Ultra valve (Edwards Lifesciences LLC, Irvine, CA, USA) or Portico/Navitor (Abbott Structural Heart, Santa Clara, CA, USA) allow for deeper implantation resulting in less challenging coronary access. Therefore, CA, as well as a basic knowledge of THV devices, and ideally an understanding of patient-specific anatomical variations are relevant to facilitate coronary access after TAVI and reduce contrast use and procedural time (Figure 3).

### 4.2. CMA and Implications for Valve Haemodynamics

The interaction of the native aortic valve and root with THV orientation interferes with haemodynamics and fluid flow behaviour. Commissural misalignment has been associated with increased leaflet stress and reduction of flow velocity within the sinuses of Valsalva and might alter the flow in the vicinity of the native commissures, thus predisposing it to blood stagnation and thrombogenicity, promoting thrombus formation (predominately within 2 months following TAVI implantation). This can result in potential adverse events such as coronary obstruction, cerebrovascular events, and early valve deterioration [5,19]. This deterioration may be due to an increase in the incidence of subclinical leaflet thrombosis, which has been reported more frequently when CMA has occurred [20,21]. Some factors, such as stent frame expansion and fracture, depth implantation, and neo-commissure symmetry and orientation, are involved in re-endothelialisation after TAVR. Thus, severe commissural misalignment may contribute to endothelial damage and dysfunction, with the subsequent delay in re-endothelialisation increasing the risk of leaflet thrombosis being a consistent explanation of these increased subclinical thrombotic events [22,23,24]. Further clinical evidence is needed to evaluate this assumption.

The evidence regarding haemodynamic outcomes and CMA remains controversial. Tirado-Conte et al. reported an association between severe CMA and an increase in post-TAVI mean aortic gradients in patients with the Portico valve [10]. Similarly, in the RESOLVE trial, CMA was associated with an increase in the mean aortic gradient at 30 days post-TAVI; however, these results have not been corroborated [4,23]. Further studies are needed to assess the haemodynamic implications of CMA, as well as the long-term impact on valve durability.

### 4.3. CA and Redo-TAVI

TAVI-in-TAVI procedures have an inherent complexity and might lead to specific complications such as sinus sequestration and coronary obstruction, related to the neo-skirt created with leaflets of the initial THV and the stent of the second THV. In bioprostheses with adequate CA, leaflet splitting with either a guide or a specific device is an effective technique to prevent iatrogenic coronary artery obstruction [25,26]. However, moderate-to-severe CMA or coronary overlap precludes effective leaflet laceration as the open space is not adjacent to the coronary ostium. Another interventional strategy with acceptable short-term results to treat coronary artery obstruction is the implantation of a coronary chimney stenting, which has been used as a preventive or bail-out technique to avoid coronary obstruction by extending a stent from the coronary ostium cranially and externally parallel to the transcatheter heart valve [27]. Future evidence with long-term clinical follow-up is needed to evaluate the safety and effectiveness of this procedure. Again, the possibility of coronary obstruction in TAVI-in-TAVI procedures reinforces the importance of adequate CA, especially in young patients with a higher probability of a subsequent aortic re-intervention [28,29].

## 5. How to Ensure Commissural Alignment

Since it has come to light that CA is an important part of a successful TAVI procedure, a partnership between structural interventional cardiologists and device companies has sought to develop intuitive and reproducible techniques to ensure alignment of neo-commissures in contemporary THVs. Procedural planning with a pre-TAVI CTA is a mandatory step in the feasibility of performing a successful CA. A CTA enables calculation of the optimal C-arm angulation to align the sinuses in 3-cusp and left/right cusp-overlap views, to minimise the parallax of the THV frame during implantation, and to infer the correct location of the coronary ostia (Figure 4). In addition, all THVs have markers within the stent frame, which represent the valve commissure position, and can be easily identified with fluoroscopy. These markers vary between different bioprostheses (Figure 5) and can assist with the evaluation of CA after valve deployment. The CA techniques are based on THV port orientation during the catheter insertion or advancing and identifying the commissure posts’ markers in a pre-determined view that varies between THV devices. THV-specific techniques are detailed below.

### 5.1. Balloon-Expandable Prostheses

Current generations of balloon-expandable prostheses have neo-commissural markers as radiopaque tabs between the hexagonal crowns identifiable only when the prosthesis is expanded. Given this, no CA technique is currently validated. Different commissural orientations during crimping have been tested with the SAPIEN 3 device (Edwards Lifescience LLC, Irvine, CA, USA), with the Edwards logo facing upwards at 12 o’clock to evaluate the position with major commissure alignment. Although the commissure crimped at 3 o’clock seems to have less severe coronary overlap, no statistical significance was observed between different crimped orientations [30]. Alternative CA techniques have been developed based on CTA analysis with dedicated software to calculate the patient-specific angle of commissure crimping to obtain adequate CA, with promising initial results that require further investigation in larger populations [31]. Importantly, the next generation SAPIEN X4 (Edwards Lifescience LLC, Irvine, CA, USA) has a specific marker to assess the orientation of the THV once crimped and positioned in the aortic plane and has a feature to axially rotate to obtain a correct CA prior to valve deployment [32].

To evaluate CA after valve deployment, the gold standard is post-TAVI CTA, but this is not routinely performed. Spilias et al. described a methodology of CA assessment specifically for SAPIEN 3 (Edwards Lifescience LLC, Irvine, CA, USA), with overlaid final aortograms and three-dimensional pre-TAVI CTA images counting the number of struts between the native and prosthetic commissures, thus calculating the CA angle taking count of each strut representing an angle of 30°. This technique strongly correlates with post-TAVI CTA, with an absolute mean angle difference of 5.1 ± 3.9° [33]. Coronary overlap can be assessed with final aortography only in the left/right cusp-overlap view, achieving acceptable CA when one commissure post is isolated on the right side of the aortic root. This assessment applies to all the THV devices and represents a CA quick check.

### 5.2. Self-Expandable Prostheses

#### 5.2.1. Evolut R/Pro and FX (Medtronic)

The markers that determine adequate CA in this THV is the C-paddle and the ‘hat’ marker. One of the commissures is oriented in the same direction as the C-paddle that is mounted 90° clockwise from the fluoroscopic ‘hat’ marker. Therefore, a centre-front position of the hat marker in the right/left cusp-overlap view places the C-paddle to the right of the screen, in the same direction as the native right/left commissure. Insertion of the delivery system is performed with the delivery system flush port at 3 o’clock. In the cusp-overlap view, the hat marker should be oriented at the centre-front during valve implantation, with the C-paddle to the right side of the image (Figure 6). The orientation of the delivery system port at 3 o’clock resulted in a reduction of severe CMA, with a left coronary CO in 15%, RCA in 7–11%, both coronaries in 2.5–5.4%, and any coronary in 19% [30,34]. A new-generation Medtronic self-expandable THV, the Evolut FX system includes three radiopaque gold markers to improve positioning accuracy and commissural alignment (Figure 6). The first-in-human experience published recently reached a 93% CA versus 80% obtained by the Evolut Pro system [35].

#### 5.2.2. Acurate Neo (Boston Scientific)

This valve has three radiopaque commissural posts and ‘free stent struts’ in line with the THV commissures (Figure 7). It is recommended to orientate the position of the delivery system flush port at 6 o’clock to reach an optimal alignment in 80% of cases [36]. Once the delivery system is advanced above the aortic annulus, the position of the commissural post can be verified. In the left/right overlap view, one commissural post is isolated to the right of the screen, and two posts are to the left with a 60° rotation. Moving the C-arm to the coplanar view, one post is at the centre-back and two posts at both sides with an angle of 60°. Additionally, only one of the free stent struts should be facing the right of the screen in the left/right overlap view. If the markers are not in the correct place in the coplanar view (a I–I–I configuration of the posts), clockwise rotation (if configuration is II–I) and anti-clockwise (if configuration is I–II) can be applied to align the radiopaque markers and reach an optimal CA in almost all cases [36,37]. It is recommended to rotate when the radiopaque marker is 5–10 mm above the aortic annulus. It is also important to understand that the torque applied is not fully conducted at the end of the system, and a torque build-up should be performed to reach the desired orientation, with a final step of releasing the system to remove the over-torque tension and avoid further rotation during implantation [19]. The isolated post and free stent strut to the right should result in THV implantation with a neo-CA.

Techniques dedicated to perform a patient-specific orientation of the delivery system insertion based on CTA analysis with dedicated software have promising preliminary results, and an ongoing clinical trial may validate this as an additional option to ensure commissure alignment [38].

#### 5.2.3. Portico/Navitor (Abbott Structural Heart)

These THVs have three radiopaque commissural posts that can be used to assess orientation of the neo-commissures and ensure appropriate alignment. Whilst no universal recommendation regarding the FlexNav delivery system orientation has been made, reports have mentioned orientating the flush port of the FlexNav delivery system at 12 o’clock when introducing the THV in order to optimise the chances of obtaining CA [6], but evidence published by Tirado-Conte et al. identified moderate-to-severe CMA in 68% of cases using this method [10]. This was as a result of the retainer receptacles used to attach the THV not being placed consistently on the FlexNav delivery system. At this moment, best practice is to insert the FlexNav delivery system catheter with port orientation at 12 o’clock, advance the THV just above the native aortic annulus, and verify the commissural post orientation. One of the posts must be isolated and aligned at the right of the screen in the right/left cusp-overlap view, with a port configuration of I–I–I in the coplanar view in order to achieve optimal CA. If the commissural posts are not well-orientated, fine-tuning of the commissural posts by rotation of the flexible FlexNav delivery system should be undertaken to reach the recommended post orientation. Clinical practice and some evidence suggest that if an isolated post is on the left side of the screen in the coplanar view, an anti-clockwise rotation can minimise the rotation degree to optimise CA, and if isolated on the right side, a clockwise rotation can be used [39]. As previously stated, the applied torque to the system is not fully conducted, and a torque build-up should be performed to reach the desired orientation, following which it is important to remember to release the system to remove the over-torque tension and avoid further rotation during implantation (Figure 8).

A promising patient-specific technique based on CTA analysis using proprietary software is currently under development, which can identify the view where the commissure posts follow the rotation of the aorta, thus maintaining alignment.

The COMALIGN trial assessed three techniques with Evolut R/Pro, Acurate Neo, and Portico/Navitor in 60 consecutive patients with systematic rotation of the delivery system to position the commissure marker. Although this manoeuvre has not been validated in larger populations, and the TAVI operators have expressed concern regarding delivery system damage or THV detachment, it was feasible and safe in 95% of cases in this particular study, obtaining implantations with ≤mild CMA in 88% of patients [6]. These findings suggest that gentle delivery system rotation can be performed without major complications. Further clinical evidence is required to support this concept.

## 6. Bicuspid Aortic Valve and Cusp Asymmetry

In patients with type 0 bicuspid aortic valves, cusp asymmetry, or coronary eccentricity, the left/right cusp-overlap view cannot accurately infer the location of the native commissures. In these scenarios, coronary alignment is preferred. As previously mentioned, CTA analysis can be utilised to objectively identify optimal projections where the left and right coronary ostia overlap, with isolation of the post to the right of the screen, which represents the furthest point from both ostia (Figure 9). This patient-specific approach is increasingly used to reduce coronary overlap and coronary access difficulty, whilst maintaining the option to use leaflet laceration techniques in future valve-in-valve procedures.

## 7. Future Perspectives

Commissural alignment in TAVR will likely be a mandatory step of every single procedure in the near future. Newer-generation valves will incorporate specific features for a better achievement of commissural alignment. Radiopaque markers in the bottom of the stent frame of the new Evolut FX system (Medtronic) and the upcoming Navitor new generation (Abbott) will facilitate orientation and improve precision on the implantation technique. The new Sapien X4 (Edwards Lifesciences, Irvine, CA, USA) with the “C” marker in the delivery system allows not only knowing the exact orientation of the valve in the annulus plane but also rotating the valve before implantation if necessary to achieve a better alignment. Sophisticated techniques based on CT imaging utilization to personalize angle device insertion and deployment to corroborate commissure alignment are being evaluated lately with promising preliminary results. Future publications with these techniques and its final results analysis will be available soon. Finally, TAVR operators will incorporate the effective ones. In addition, the TAVR implanters should be involved in discovering new implantation techniques to avoid a moderate-to-severe CMA.

## 8. Conclusions

CAs in TAVIs can result in easier coronary access and better coronary artery flow/filling and valve haemodynamics, and they can improve the function of the active outer sealing skirt. Furthermore, CA maintains the feasibility of performing leaflet laceration in cases of redo-TAVI and reduces leaflet stress and strain, thereby positively influencing the longevity of the device. Whilst the techniques described herein have been developed to improve CA, future individualised patient-specific techniques are the next step to ensure CA is achieved in almost all patients undergoing TAVI.

## Figures and Tables

**Figure 1 jcm-12-07369-f001:**
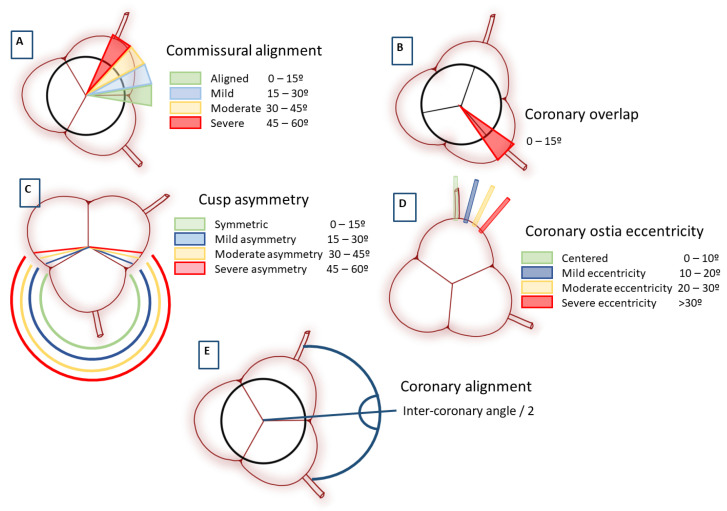
Definitions. Definitions and classifications related to Commissural misalignment (**A**), Coronary overlap (**B**), Cusp asymmetry (**C**), Coronary eccentricity (**D**), and Coronary alignment (**E**).

**Figure 2 jcm-12-07369-f002:**
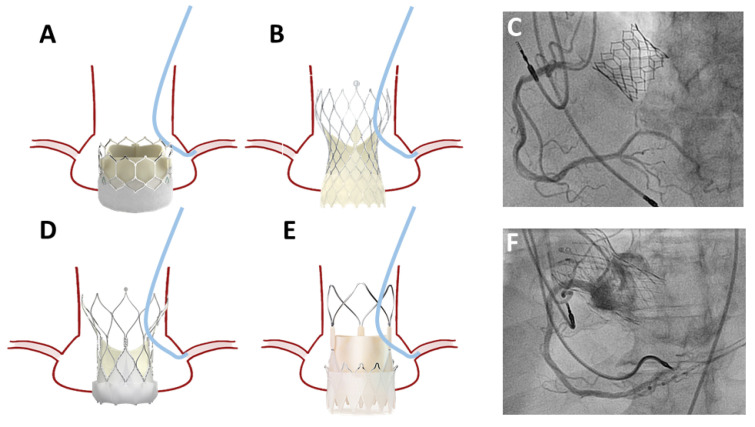
Coronary access with different TAVI devices. In balloon-expandable bioprostheses, coronary access is in the space between the device and sinotubular junction without interaction between the catheter and the device in the majority of the cases (**A**,**C**). In self-expandable bioprostheses, the coronary access is through the bioprosthetic struts, which demands catheter interaction with the device (**B**,**D**–**F**).

**Figure 3 jcm-12-07369-f003:**
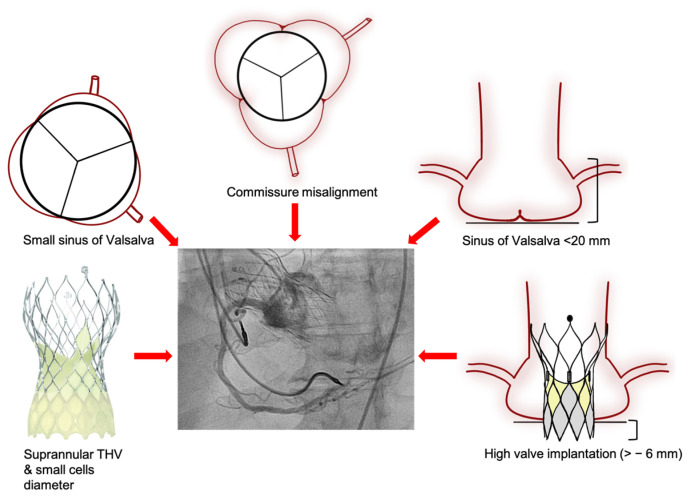
Risk factors resulting in inaccessible or semi-selective coronary cannulation after TAVI (red arrows). Risk factors that are related to an increased technical difficulty of coronary re-access following TAVI.

**Figure 4 jcm-12-07369-f004:**
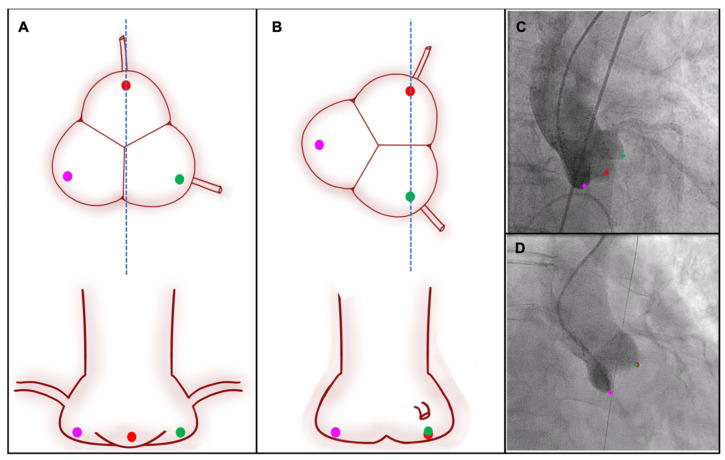
Co-planar and cusp-overlap view scheme. (**A**,**C**) represent the standard view where the 3-cusps are aligned, and the neo-commissure can be predicted. (**B**,**D**) shows the right-to-left cusp-overlap view, where the cusp between right and left cusps are isolated on the right side of the screen, enabling the alignment to be assessable. Upper images represent a transverse view of the fluoroscopic views, represented in the lower images of the panels. Red dot: right cusp, green dot: left cusp, Purple dot: non-coronary cusp. Blue dotted line: long axis view direction represented in the perpendicular plane.

**Figure 5 jcm-12-07369-f005:**
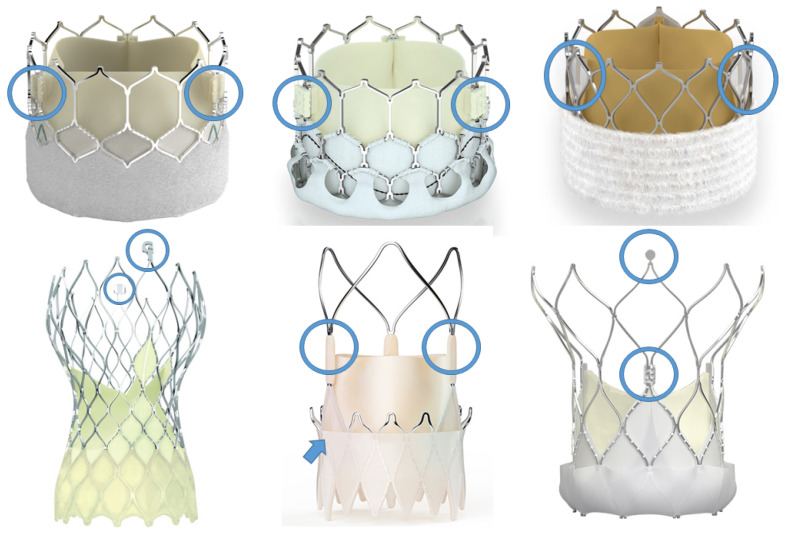
Radiopaque markers in different TAVI devices. TAVI bioprostheses have a radiopaque marker aligned within the posts where the leaflets are sutured (blue circles).

**Figure 6 jcm-12-07369-f006:**
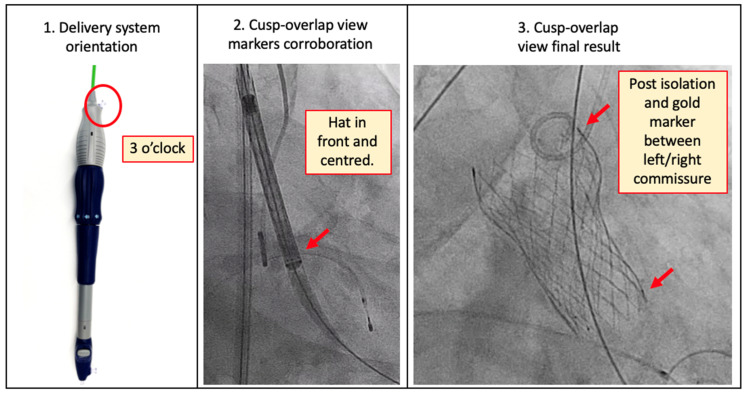
Evolut FX system and commissural alignment technique. Firstly, the delivery system is introduced with flush port (red circle) pointing towards 3 o’clock (pointing away from the operators). Secondly, corroborate in cusp-overlap view that the hat marker is in front and centred. Finally, after implantation, evaluate in the same view, and ensure that the C-paddle and one radiopaque marker are isolated to the right of the screen (red arrows).

**Figure 7 jcm-12-07369-f007:**
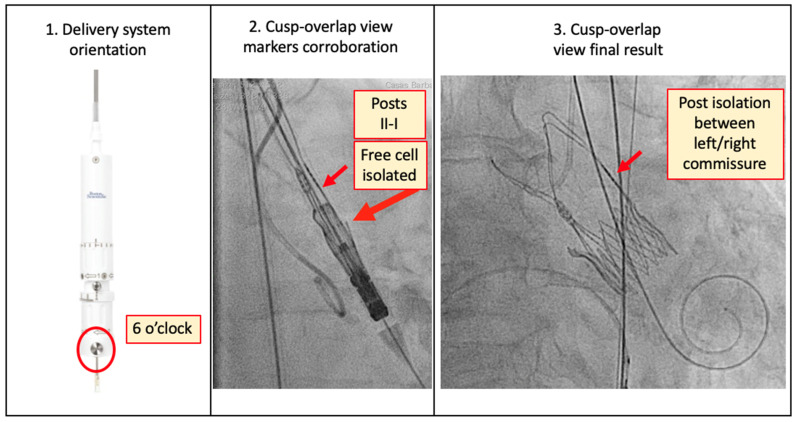
Acurate neo system and commissural alignment technique. Firstly, the delivery system is introduced with the flush port (red circle) pointing towards 6 o’clock. Secondly, corroborate with the cusp-overlap view that the free cell and a post marker are isolated to the right of the screen (represented as a black line on the fluoroscopy, and two posts to the left are represented as two overlapping rectangles). Finally, after implantation, in the same view, evaluate that one radiopaque marker is isolated to the right of the screen (red arrows).

**Figure 8 jcm-12-07369-f008:**
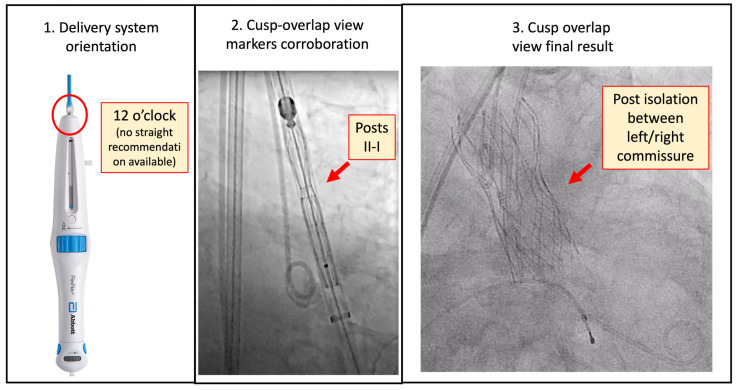
Portico/Navitor system and commissural alignment technique. Firstly, the delivery system is introduced with the flush port (red circle) at 12 o′clock, as recommended by the developers. Secondly, corroborate in cusp-overlap view that the post marker is isolated to the right of the screen (represented as a black line in the fluoroscopy; two posts on the left represented as two rectangles overlapped). Finally, after implantation, evaluate in the same view that one radiopaque marker is isolated to the right of the screen (red arrows).

**Figure 9 jcm-12-07369-f009:**
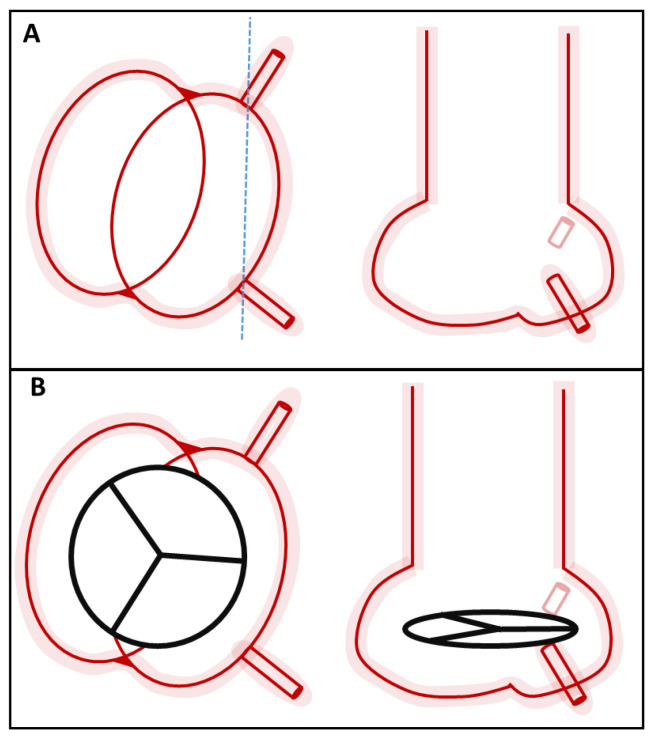
Coronary alignment in bicuspid aortic valve. (**A**) shows the inter-coronary aligned view (blue line) and the respective view along the long axis that places the longest distance from both coronary ostia isolated to the right of the screen. (**B**) represents an example when the TAVI is implanted with a radiopaque marker isolated to the right using this view.

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
