# Peer review of "The Incidence, Impact, and Techniques of Commissural Alignment in Transcatheter Aortic Valve Implantation: A Review"

_jcm, 2023, doi:10.3390/jcm12237369_

Round 1
Reviewer 1 Report
Comments and Suggestions for Authors
I read with great interest the review article “Incidence, impact, and technique of commissural alignment in transcatheter aortic valve implantation: a review”. Thank you for this manuscript. I am pleased to comment on it as a reviewer.
The review article discusses a topic in the field of transcatheter valves (i.e., TAVI) that is of increasing clinical importance. As TAVI is now an established therapy in high- and intermediate-risk patients, optimization of the implantation technique is more and more relevant. It is known that axial malrotation of the valve can lead to impaired coronary access, hemodynamics, or valve durability, as well as valve thrombosis. This article summarizes all current findings on this topic based on the existing literature.
The manuscript is well written and clearly organized. It contains the relevant literature on TAVI misalignment and shows ample visual material to illustrate this issue. The article begins with various definitions and illustrations of the following geometric concepts: commisural misalignment, aortic cusp asymmetry, ostial eccentricity, and coronary overlap. According to a cited study, too little attention has been paid to the correct rotation of a TAVI in contrast to surgical valve implantation, as it has been based more on chance (SAVR - 96% correct, TAVI - 22% correct).
The next chapter systematically highlights and graphically depicts the main harmful effects of commissural misalignment. Coronary access may be compromised for reasons of anatomy, valve design, and implantation height (this is especially true for self-expanding valves such as Evolut). Valve hemodynamics can also be impaired, leading to increased leaflet stress, premature degeneration, and valve thrombosis.
Finally, wide space was given to the description of the optimized implantation technique. The most important current TAVI types were considered, such as Sapien, Evolut, Acurate Neo and Portico, and also the special features of bicuspid aortic valves.
Overall, the review article gives a clear and concise overview of the topic of commissural misalignment while also providing device-specific advice for improvement. Therefore, I would recommend this article without significant changes. More data on the correct alignment of transcatheter valves will certainly be published in the future.
Please note only two minor details:
- Please check the consistency of Fig. 4 regarding the coronary arteries (upper vs. lower).
- In Fig. 6, I didn't really get the point of the “3 o'clock position”.
Thank you.
Author Response
Thank you for your review of this manuscript, definitely these improving points represents an important upgrade of this paper.
The recommendations described in the review letter where addressed below:
Review 1:
I read with great interest the review article “Incidence, impact, and technique of commissural alignment in transcatheter aortic valve implantation: a review”. Thank you for this manuscript. I am pleased to comment on it as a reviewer.
The review article discusses a topic in the field of transcatheter valves (i.e., TAVI) that is of increasing clinical importance. As TAVI is now an established therapy in high- and intermediate-risk patients, optimization of the implantation technique is more and more relevant. It is known that axial malrotation of the valve can lead to impaired coronary access, hemodynamics, or valve durability, as well as valve thrombosis. This article summarizes all current findings on this topic based on the existing literature.
The manuscript is well written and clearly organized. It contains the relevant literature on TAVI misalignment and shows ample visual material to illustrate this issue. The article begins with various definitions and illustrations of the following geometric concepts: commisural misalignment, aortic cusp asymmetry, ostial eccentricity, and coronary overlap. According to a cited study, too little attention has been paid to the correct rotation of a TAVI in contrast to surgical valve implantation, as it has been based more on chance (SAVR - 96% correct, TAVI - 22% correct).
The next chapter systematically highlights and graphically depicts the main harmful effects of commissural misalignment. Coronary access may be compromised for reasons of anatomy, valve design, and implantation height (this is especially true for self-expanding valves such as Evolut). Valve hemodynamics can also be impaired, leading to increased leaflet stress, premature degeneration, and valve thrombosis.
Finally, wide space was given to the description of the optimized implantation technique. The most important current TAVI types were considered, such as Sapien, Evolut, Acurate Neo and Portico, and also the special features of bicuspid aortic valves.
Overall, the review article gives a clear and concise overview of the topic of commissural misalignment while also providing device-specific advice for improvement. Therefore, I would recommend this article without significant changes. More data on the correct alignment of transcatheter valves will certainly be published in the future.
We thank reviewer 1 for his/her positive comments and we appreciated the inputs provided.
Suggestion:
- Please check the consistency of Fig. 4 regarding the coronary arteries (upper vs. lower).
We thank to the reviewer for pointing out this mistake. Accordingly we have modified figure 1 in relation to the take of the coronary arteries.
Suggestion:
- In Fig. 6, I didn't really get the point of the “3 o'clock position”.
The flush port of the delivery system was usually introduced at 12 o´clock (pointing up). One quarter clock-wise rotation of the delivery system has been recommended for Evolut valve implantation, so the flush port at 3 o´clock (pointing away from the operators). We have included this information in the legend of figure 6.
Reviewer 2 Report
Comments and Suggestions for Authors
Dear authors, this is a very interesting and sophisticated review covering all important aspects of the topic. Some questions could be adressed to even further increase the quality of the manuscript.
- does CA also play a role concerning endothelial dysfunction and/or hemolysis
- What are your thoughts concerning longevity of TAVI valves taking CA into account?
- Please give a short statement concerning chimney technique as coronary protection option in chapter 4.3.?
- Could you integrate the 3 gold markers of the Evolut FX in figure 5 and the new FX implantation strategy in chapter 5.2.1?
- Could you add a chapter "future perspectives"?
Author Response
Thank you for your review of this manuscript, definitely these improving points represents an important upgrade of this paper.
The recommendations described in the review letter where addressed below:
Dear authors, this is a very interesting and sophisticated review covering all important aspects of the topic. Some questions could be adressed to even further increase the quality of the manuscript.
Suggestion:
- Does CA also play a role concerning endothelial dysfunction and/or hemolysis
We thanks the reviewer for this input. We have added a comment in the section “CMA and implications on valve haemodynamics” a relationship between the endothelial dysfunction and the CMA as follow: “Some factors as stent frame expansion and fracture, depth implantation and neo-commissures symmetry and orientation are involved in the re-endothelization after TAVR. So, severe commissural misalignment may contribute to endothelial damage and dysfunction with the subsequent delay in the re-endothelization increasing the risk of leaflet thrombosis being a consistent explanation of this increased subclinical thrombotic events”.
Suggestion:
- What are your thoughts concerning longevity of TAVI valves taking CA into account?
Durability is the major concern about the CMA because represents future high-risk reinterventions in patients increasingly youngers, hospitalizations and cardiovascular events. Unfortunately, no large and conclusive data is available nowadays. There are some studies suggesting that valves with no or mild CMA had better valve hemodynamics that valves with severe CMA. It is clear that lower gradients through the valve will have an impact in long-term outcomes and durability. However there is no data to prove that patients with severe CMA have greater valve deterioration. Addressing this lacking of evidence about it in our manuscript, we mentioned as follow: “Further studies are needed to assess the haemodynamic implications of CMA, as well as the long-term impact on valve durability”.
Suggestion:
- Please give a short statement concerning chimney technique as coronary protection option in chapter 4.3.?
We totally agree with the reviewer. As suggested, we added in the section the paragraph mentioned below with an additional reference about the cardiac tomography findings in patients undergoing chimney technique as coronary protection.
The following sentences have been added in the newer version of the manuscript.
“Another interventional strategy with acceptable short-term results to treat coronary artery obstruction is the implantation of a coronary chimney stenting, that has been used as a preventive or bail-out technique to avoid coronary obstruction by extending a stent from the coronary ostium cranially and externally parallel to the transcatheter heart valve. Future evidence with long-term clinical follow-up is needed to evaluate the safety and effectiveness of this procedure”.
Suggestion:
- Could you integrate the 3 gold markers of the Evolut FX in figure 5 and the new FX implantation strategy in chapter 5.2.1?
We thank the reviewer for this comment. We have incorporated the new Evolut-FX in figure 5 and added in the section the following data: “New generation Medtronic self-expandable THV, the Evolut FX system includes three radiopaque gold markers to improve positioning accuracy and commissural alignment. The first-in-human experience published recently reached a 93% of CA against 80% obtained by Evolut Pro system”.
Suggestion:
- Could you add a chapter "future perspectives"?
We added a chapter with the future perspectives before the conclusions.